# VALUE PROPAGATION NETWORKS

## ABSTRACT

We present Value Propagation (VProp), a parameter-efficient differentiable planning module built on Value Iteration which can successfully be trained in a reinforcement learning fashion to solve unseen tasks, has the capability to generalize to larger map sizes, and can learn to navigate in dynamic environments. We evaluate on configurations of MazeBase grid-worlds, with randomly generated environments of several different sizes. Furthermore, we show that the module enables to learn to plan when the environment also includes stochastic elements, providing a cost-efficient learning system to build low-level size-invariant planners for a variety of interactive navigation problems.

## 1    INTRODUCTION

Planning is a key component for artificial agents in a variety of domains. However, a limit of classical planning algorithms is that one needs to know how to search for an optimal (or good) solution, for each type of plan. When the complexity of the planning environment and the diversity of tasks increase, this makes writing planners difficult, cumbersome, or entirely infeasible. "Learning to plan" has been an active research area to address this shortcoming (Russell et al., 1995; Kaelbling et al., 1996). To be useful in practice, we propose that methods for learning to plan should have at least two properties: they should be *traces free*, i.e. not require traces from an optimal planner, and they should *generalize*, i.e. learn planners that generalize to plans of the same type but of unseen instance and/or planning horizons.

In a Reinforcement Learning (RL) setting, learning to plan can be framed as the problem of finding a policy that maximises the expected reward, where such policy is a greedy function that selects actions that will visit states with a higher value for the agent. In such cases, Value Iteration (VI) is a algorithm that is naturally used to learn to estimate the value of states, by propagating the rewards and values until a fixed point is reached. When the environment can be represented as an occupancy map (a 2D grid), it is possible to approximate this learning algorithm using a deep convolutional neural network (CNN) to propagate the value on the grid cells. This enables one to differentiate directly through the planner steps and perform end-to-end learning. One way to train such models is with a supervised loss on the trace from a search/planning algorithm, e.g. as seen in the supervised learning section of Value Iteration Networks (VIN) (Tamar et al., 2016), in which the model is tasked with reproducing the function to iteratively build values aimed at solving the shortest path task. However, this baseline violates our wished trace free property because of the required target values, and it doesn't fully demonstrate the capabilities to deal with interactive and generalized settings. That is what we set out to extend and further study.

In this work we extend the formalization used in VIN to more accurately represent the structure of grid-world-like scenarios, enabling Value Iteration modules to be naturally used within the reinforcement learning framework, while also removing some of the limitations and underlying assumptions of the model. Furthermore we propose hierarchical extensions of such a model that allow agents to do multi-step planning, effectively learning models with the capacity to provide useful path-finding and planning capabilities in relatively complex tasks and comparably large scenarios. We show that our models can not only learn to plan and navigate in complex and dynamic environments, but that their hierarchical structure provides a way to generalize to navigation tasks where the required planning and the size of the map are much larger than the ones seen at training time.

Our main contributions include: (1) introducing VProp, a network module which successfully learns to solve pathfinding via reinforcement learning, (2) demonstrating the ability to generalize, leading

our models to solve large unseen maps by training exclusively on much smaller ones, and (3) showing that our modules can learn to navigate environments with more complex dynamics than a static grid-world.

## 2 BACKGROUND

We consider the control of an agent in a "grid world" environment, in which entities can interact with each other. The entities have some set of attributes, including a uniquely defined type, which describes how they interact with each other, the immediate rewards of such interactions, and how such interactions affect the next state of the world. The goal is to *learn to plan* through reinforcement learning, that is learning a policy trained on various configurations of the environment that can generalize to arbitrary other configurations of the environment, including larger environments, and ones with a larger number of entities. In the case of a standard navigation task, this boils down to learning a policy which, given an observation of the world, will output actions that take the agent to the goal as quickly as possible. An agent may observe such environments as 2D images of size $d_\mathrm{x} \times d_\mathrm{y}$, with $d_\mathrm{pix}$ input panes, which are then potentially passed through a local embedding function $\Phi : \mathbb{R}^{d_\mathrm{pix} \times d_\mathrm{x} \times d_\mathrm{y}} \to \mathbb{R}^{d_\mathrm{rew} \times d_\mathrm{x} \times d_\mathrm{y}}$. The function could for instance be implemented as a 2D convolutional network that extracts the type and position of entities.

This formalisation of the navigation problem is often employed in robotics to represent 3D and 2D surfaces, finding its use in, but not limited to, frameworks such as Simultaneous Localization and Mapping (SLAM), physics simulators, and more generic planning environments. This makes it appropriate for developing models and evaluating their impact on real-world planning tasks (Thrun et al., 2005).

### 2.1 REINFORCEMENT LEARNING

The problem of reinforcement learning is typically formulated in terms of the computation of optimal policies for a Markov Decision Problem (MDP) (Sutton & Barto, 1998). An MDP is defined by the tuple $(S, A, T, R, \gamma)$, where S is a finite set of states, $A$ is the set of actions $a$ that the agent can take, $T : s \to a \to s'$ is a function describing the state-transition matrix, $R$ is a reward function, and $\gamma$ is a discount factor. In this setting, an optimal policy $\pi^*$ is a distribution over the state-action space that maximises in expectation the discounted sum of rewards $\sum_k \gamma^k r_k$, where $r_k$ is the single-step reward. A standard method (such as the one used in Q-Learning (Watkins & Dayan, 1992)) to find the optimal policy $\pi : s \to a$ is to iteratively compute the value function, $Q^\pi(s, a)$, updating it based on rewards received from the environment. Using this framework, we can view learning to plan as a *structured prediction* problem with the planning algorithm Value Iteration (Bertsekas, 2012) as inference procedure. When doing so, the problem generalises to learning a model of the environment that is approximating the state-transition probabilities and the immediate reward function. Beside value-based algorithms, there exist other types which are able to find optimal policies, such as *policy gradient* methods, which directly regress to the policy function $\pi$, instead of approximating the value function. Finally, a third type is represented by the actor-critic algorithms family, which combine the policy gradient methods' advantage of being able to compute the policy directly, with the low-variance performance estimation of value-based RL used as a more accurate feedback signal to the policy estimator (Konda & Tsitsiklis, 2000).

### 2.2 VALUE ITERATION MODULE

Let us denote by $o$ the current observation of the environment, and $q^0$ the zero tensor of dimensions $(A, d_\mathrm{x}, d_\mathrm{y})$. Then, the *Value Iteration* (VI) module proposed by Tamar et al. (2016) is a recurrent neural network where each layer first performs max-pooling over the first dimension followed by a layer $h : \left( \mathbb{R}^{d_\mathrm{rew} \times d_\mathrm{x} \times d_\mathrm{y}} \right) \times \left( \mathbb{R}^{d_\mathrm{x} \times d_\mathrm{y}} \right) \to \mathbb{R}^{A \times d_\mathrm{x} \times d_\mathrm{y}}$, giving rise to the following recurrence, for $k \geq 1$:

$$\forall (i,j) \in [\![ d_\mathrm{x} ]\!] \times [\![ d_\mathrm{y} ]\!], \; v_{ij}^k = \max_{a=1..A} q_{aij}^k, \; q^k = h(\Phi(o), v^{k-1}). \tag{1}$$

Given the agent's position $(x_0, y_0)$ and the current observation of the environment $o$, the control policy $\pi$ is then defined by $\pi(o, (x_0, y_0)) = \mathrm{argmax}_{a=1..A} \, q_{ax_0 y_0}$.

Since the state space can be identified with the coordinates in a 2-dimensional environment of size $d_{\mathrm{x}} \times d_{\mathrm{y}}$ then a discounted MDP on that state is defined by the discount factor $\gamma$ together with:

- a transition probability function $T := d_{\mathrm{x}} \times d_{\mathrm{y}} \times A \times d_{\mathrm{x}} \times d_{\mathrm{y}}$, such that the matrix obtained fixing the last three coordinates sum to 1.

- an immediate reward function $R := A \times d_{\mathrm{x}} \times d_{\mathrm{y}}$, where $R_{a,i,j}$ represents the reward obtained by performing action $a$ in state $i, j$.

Given a starting $Q$-function $Q^0$ (with same size as $R$), value iteration defines a sequence $Q^k, V^k$ of (state, action)- and state-value functions through:

$$\forall (i,j) \in [\![\, d_{\mathrm{x}} \,]\!] \times [\![\, d_{\mathrm{y}} \,]\!], \;\; V_{ij}^k = \max_{a=1..A} Q_{aij}^k$$

$$\forall (a,i,j) \in [\![\, A \,]\!] \times [\![\, d_{\mathrm{x}} \,]\!] \times [\![\, d_{\mathrm{y}} \,]\!], \;\; Q_{aij}^k = R_{aij} + \gamma \langle T_{::aij}, V^{k-1} \rangle \,.$$

It follows that when $d_{\mathrm{rew}} := A$ and $\Phi(x) := R$, $h$ is a linear layer of both of its inputs with $T$ as parameters for the $v^k$ input:

$$h_{aij}(\Phi(o), v) := \Phi_{aij}(o) + \langle \gamma T_{::aij}, v \rangle. \tag{2}$$

Thus, a linear VI module with $K$ iteration has the capacity to represent the application of $K$ iterations of value iteration in an MDP where the state space is the set of 2D coordinates, requiring in the worst case a number of steps equal to the number of states – $d_{\mathrm{x}} d_{\mathrm{y}}$ in grid worlds – to evaluate an entire plan. In practice however, much fewer iterations are needed in the considered tasks: for instance path finding in a grid-world requires planning for only the length of the shortest path, which is much smaller unless the configuration corresponds to a very complicated (and unusual) maze.

We can also see that (2) corresponds to only a very special case of linear VI modules. For instance, if the recurrence uses a fully connected layer with weights $W \in \mathbb{R}^{A \times d_{\mathrm{x}} \times d_{\mathrm{y}} \times (d_{\mathrm{rew}}+1) \times d_{\mathrm{x}} \times d_{\mathrm{y}}}$ and biases $b \in \mathbb{R}^{A \times d_{\mathrm{x}} \times d_{\mathrm{y}}}$ so that

$$h_{aij}(\Phi(o), v) = \langle W_{aij}, [\Phi(o)\,; v] \rangle + b_{aij} \,,$$

then (2) not only corresponds to a special case with extremely sparse weights, but also exhibits a specific structure where the dot product is non-trivial only on the recurrent part. This relationship ultimately forms the motivation for the development of VI modules (Tamar et al., 2016, Section 3.1).

## 3 Related work

Model-based planning with end-to-end architectures has recently shown promising results on a variety of tasks and environments, often using Deep Reinforcement Learning as the algorithmic framework (Silver et al., 2016; Oh et al., 2017; Weber et al., 2017; Groshev et al., 2017). 3D navigation tasks have also been tackled within the RL framework (Mirowski et al., 2016), with methods in some cases building and conditioning on 2D occupancy maps to aid the process of localization and feature grounding (Bhatti et al., 2016; Zhang et al., 2017).

Other work has furthermore explored the usage of VIN-like architectures for navigation problems: Niu et al. (2017) present a generalization of VIN able to learn modules on more generic graph structures by employing a graph convolutional operator to convolve through each node of the graph. Rehder et al. (2017) demonstrate a method for multi-agent planning in a cooperative setting by training multiple VI modules and composing them into one network, while also adding an orientation state channel to simulate non-holonomic constraints often found in mobile robotics. Gupta et al. (2017) and Khan et al. (2017) propose to tackle partially observable settings by constructing hierarchical planners that use VI modules in a multi-scale fashion to generate plans and condition the model's belief state.

It is clear that all such work can indeed greatly benefit from employing a powerful planner with capable of learning in a larger combination of tasks and environments, motivating us to develop one which can directly substitute VIN.

# 4 MODELS

In this section we formalize two novel alternatives to the Value Iteration module, both of which can function as drop-in replacements for VIN and be used as low-level planners in graph-based tasks.

## 4.1 VALUE-PROPAGATION MODULE

A problematic and key limitation of the VI module (1) is that the transition probabilities are encoded as the weights $W$ of the convolution, which are naturally translation-invariant. This means that the transition probabilities at one state, a cell in this case, do not depend on its content or the surrounding ones'. This restriction affects the capability of the model to learn any complex dynamics, while also artificially constraining the design of the architecture and other experimental parameters. For instance, non-achievable state must be implemented so that they can be represented by a sufficiently low cost to compensate for any surrounding positive cost.

We can obtain a minimal parametrization of a VI module that represents reasonable prior knowledge on the model by simply observing that the transition probabilities must depend on the state we are in, and that the rewards associated to transitions from state $s$ to $s'$ with action $a$ takes an additive form $r_{s'}^{\text{in}} - r_s^{\text{out}}$, where $r^{\text{in}}$ and $r^{\text{out}}$ depend on the state at hand.[1]

We can now therefore consider a variant of the VI module that puts emphasis on conditioning on a different aspect of the grid's environment dynamics: state reachability. Our new module, which we call *Value Propagation* module (VProp), takes as input three embeddings: a *value* $r^{\text{in}}(o)$, a *cost* $r^{\text{out}}(o)$, and a *propagation* $p(o)$, all of which are scalar maps of size $\mathbb{R}^{d_\text{x} \times d_\text{y}}$. Using these functions as input, the resulting output is then computed as

$$v^{(0)} = \mathbf{0}_{d_\text{x} \times d_\text{y}}, \text{ and for } k \in \{1, ..., K\}, v_{i,j}^{(k)} = \max\left(v_{i,j}^{(k-1)}, \max_{(i',j') \in \mathcal{N}(i,j)} \left(p_{i,j} v_{i',j'}^{(k-1)} + r_{i',j'}^{\text{in}} - r_{i,j}^{\text{out}}\right)\right),$$

where $v^{(k)}$, $r^{\text{in}}$, $r^{\text{out}}$ and $p$ depend on the observation $o$, and $\mathcal{N}(i,j)$ represents the coordinates of the cells adjacent to the cell $(i,j)$. Note that here we made the convolution and the maximum explicit, since they can be done in a single step.

VProp's model corresponds to a deterministic model in which the reward propagates from adjacent states to current states, capturing the prior that the reachability of a state does not depend on the adjacent cell the agent is in, but by the underlying – and less constraining – observed transition dynamics. As such, instead of setting the probability to get to a unachievable state to $0$ from its adjacent cells, the model represents unachievable states by setting their propagation to $0$ and $r^{\text{in}}$ to some negative value.[2] Goal states, which are also absorbing for the underlying MDP, are represented with a propagation close $0$, and a positive $r^{\text{in}}$, whereas other types of cell are to have high propagation value bounded by the discount factor, while the cost of their traversal is represented by either a negative $r^{\text{in}}$ or a positive $r^{\text{out}}$. Finally, given the agent's coordinates $s$, the agent policy becomes a function that takes as input the vector $[p_s v_{i',j'}^{(K)} + r_{i',j'}^{\text{in}} - r_s^{\text{out}}]_{(i',j') \in \mathcal{N}(s)}$, and either outputs an action or a distribution over actions.

## 4.2 MAX-PROPAGATION MODULE

Since both VI module and VProp module described above are purely convolutional, they should be able to exploit the lack of size constraints in the input to generalize to larger environments than seen during training time by simply increasing the recursion depth. However, despite the convolutions effectively hard-coding the local planning algorithm, these models cannot be proven to generalize properly across the entire distribution of environment configurations. To see this, let us consider the simple problem of pathfinding. Even in a simple grid-world with blocks and a single goal, there are many sets of parameters of the VI / VProp module that are equally good for pathfinding with a fixed environment size. For instance, in an environment of size $20 \times 20$ in which paths are of length $40$ with very high probability (over the randomized configurations of the environment), we can trivially check that a VProp module with a $r^{\text{in}}$ value of $1$ for the goal entity and $0$ otherwise, $r^{\text{out}}$ to $-0.02$

---

[1]Note that while we use a minus sign, we do not impose specific sign constraints on $r^{\text{in}}$ and $r^{\text{out}}$, since natural environments can easily be represented with both values as positive.

[2]Absorbing states can be represented in the same way.

for empty cells, $-1$ for blocks and $-1$ for the goal state, and with propagation set to $1$, everywhere, will effectively create a value map $v^{(K)}$ which is affinely related to the shortest path to the goal on empty cells. Thus, as far as the training distribution of environments is concerned, this would be an acceptable set of parameters. However, if we increase the size of the environment such that paths are strictly longer than 50, the goal value would become $1 - 51 \times 0.02$, which is less than $0$, hence stopping the propagation of the path after the 51st step. The model is therefore bound to the capacity of the additive rollout in the value map, which is responsible for both identifying the goal and computing the shortest path.

To solve this problem, we further introduce the Max-Value Propagation module (MVProp), which constrains the network to represent goals with a high base reward, propagates them multiplicatively through cells with lower values but high propagation, and learns reward and propagation maps depending on the content of the individual cells in the same fashion as VProp. More precisely, denoting again the initial reward map by $r(o) \in [0,1]^{d_x \times d_y}$ and the propagation map by $p(o) \in [0,1]^{d_x \times d_y}$, MVProp iteratively computes value maps $v^{(k)}(o)$ as follows:

$$v^{(0)} = r \text{ and, for } k \in \{1, ..., K\}, v_{i,j}^{(k)} = \max \left( r_{i,j}, \max_{(i',j') \in \mathcal{N}(i,j)} \left( r_{i,j} + p_{i,j}(v_{i',j'}^{(k-1)} - r_{i,j}) \right) \right).$$

The input to the policy, given the agent position $s = (i_0, j_0)$, is then the $3 \times 3$ map of $\left( v_{i',j'}^{(K)} \right)_{\substack{i' \in \{i_0-1, i_0, i_0+1\} \\ j' \in \{j_0-1, j_0, j_0+1\}}}$, padded with $0$ at the boundaries. This propagation system guarantees that values are propagated in the direction of lower-value cells at all times (even at the start of training), and that costs of individual cells are dealt with the $1 - p$ map. In other words, the path length is propagated multiplicatively, whereas the reward map is used to distinguish between goal cells and other cells. Given this setup, the optimal policy should therefore be able to locally follow the direction of maximum value.

## 5 EXPERIMENTS

Tamar et al. (2016) describes a variety of experiments to test VIN via Imitation Learning (IL), while we focus on evaluating our our modules strictly using Reinforcement Learning, since we are interested in moving towards environments that better simulate tasks requiring interaction with the environment, hence limiting the chance to gather representative datasets.

### 5.1 AGENT SETUP

For training, we use an actor-critic architecture with experience replay. We collect transition traces of the form $(s^t, a^t, r^t, p^t, s^{t+1})$, where $s^t$ is the state at time step $t$, $a^t$ is the action that was chosen, $p^t$ is the vector of probabilities of actions as given by the policy, and $r^t$ is the immediate reward. The state is represented by the coordinates of the agent and 2D environment observation, excluding any agent channel. Terminal states are instead represented by some value $\emptyset$ when sampled as $s^{t+1}$. The architecture contains the policy $\pi_\theta$ described in the previous sections, together with a value function $V_w$, which takes the same input as the softmax layer of the policy, concatenated with the $3x3$ neighborhood of the agent. Note that $w$ and $\theta$ share all their weights until the end of the convolutional recurrence. At training time, given the stochastic policy at time step $t$ denoted by $\pi_{\theta^t}$, we sample a minibatch of $B$ transitions, denoted $\mathcal{B}$, uniformly at random from the last $L$ transitions, and perform gradient ascent over importance-weighted rewards (more details on the constants $\eta, \eta', \lambda, C$ below):

$$\theta^{t+1} \leftarrow \theta^t + \eta \sum_{(s,a,r,p,s') \in \mathcal{B}} \min \left( \frac{\pi_{\theta^t}(s,a)}{p(a)}, C \right) \left( r + \mathbf{1}_{\{s' \neq \emptyset\}} V_{w^t}(s') - V_{\theta^t w^t}(s) \right) \left( \nabla_\theta \log \pi_\theta(s,a) \right)_{|\theta = \theta^t}$$

$$+ \lambda \sum_{(s,a,r,p,s') \in \mathcal{B}} \sum_{a'} p(a') \left( \nabla_\theta \log \pi_\theta(s,a') \right)_{|\theta = \theta^t},$$

$$w^{t+1} \leftarrow w^t + \eta' \sum_{(s,a,r,p,s') \in \mathcal{B}} \min \left( \frac{\pi_{\theta^t}(s,a)}{p(a)}, C \right) \left( (\nabla_w V_w(s))_{|w = w^t} - r - \mathbf{1}_{\{s' \neq \emptyset\}} V_{w^t}(s') \right)^2,$$

where $\mathbf{1}_{\{s' \neq \emptyset\}}$ is 1 if $s'$ is terminal and 0 otherwise.

The capped importance weights $\min\left(\frac{\pi_{\theta t}(s,a)}{p(a)}, C\right)$ are standard in off-policy policy gradient (see e.g., (Wang et al., 2016) and references therein). The capping constant ($C = 10$ in our experiments) allows control of the variance of the gradients at the expense of some bias. The second term of the update acts as a regularizer and forces the current predictions to be close enough the the ones that were made by the older model. It is supposed to play the role of TRPO-like regularizations as implemented in Wang et al. (2016), where they use probabilities of an average model instead of the previous probabilities. We observed that by keeping only the last 50000 transitions (to avoid trying to fit predictions of a bad model) worked well on our task. The update rule for the parameters of the value function also follows the standard rule for off-policy actor-critic. The learning rates $\eta$, $\lambda$ and $\eta'$ also control the relative weighting of the different objectives when the weights are shared. In practice, we use RMSProp rather than plain SGD, with relative weights $\lambda = \eta = 100.0\eta'$.

## 5.2 GRID-WORLD SETTING

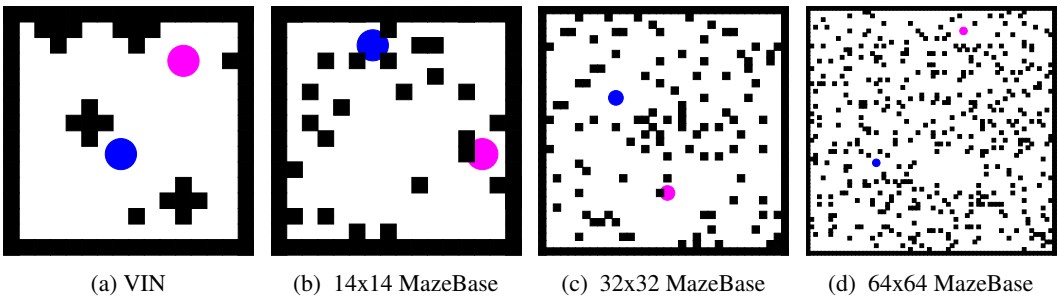

|                |                   |                   |                   |
| :------------: | :---------------: | :---------------: | :---------------: |
| (a) VIN        | (b) 14x14 MazeBase | (c) 32x32 MazeBase | (d) 64x64 MazeBase |

Figure 1: Comparison between a random map of the VIN dataset, and a few random configuration of our training environment. In our custom grid-worlds, the number of blocks increases with size, but their percentage over the total available space is kept fixed. Agent and goal are shown as circles for better visualization, however they still occupy a single cell.

Our grid-world experimental setting is mostly identical to the ones used in previous work on the same topic: an agent and a goal are placed at random positions in a 2d world of fixed dimensions. The agent is allowed to move in all 8 directions at each step, and a terminal state is reached when the agent either reaches the goal or hits one of the walls randomly placed when the task is sampled from the task distribution. The ratio of unwalkable blocks over total space is fixed to $30\%$, and the blocks are sampled uniformly within the space, unless specified otherwise (Figure 1). This setting provides environments with a good distribution of configurations, with chunks of obstacles as well as narrow and dangerous paths. Unless specified otherwise, attempting to walk into walls yields a reward of $-1$, valid movements give a reward of $-0.01 \times f(a_t, s_t)$, where $f(a, s)$ is the cost of moving in the direction specified by action $a$ in state $s$, and reaching the goal provides a positive reward of $1$ to the agent. In our experiments we define $f$ as the L2 distance between the agent position at state $s_t$, and the one at $s_{t+1}$. The agent is tasked to navigate the maze as fast as possible, as total cost increases with time since $noop$ actions are not allowed. Episodes terminate whenever the agent takes any illegal action such as hitting a wall, or when the maximum number of steps is reached (which is set to 50). We use MazeBase (Sukhbaatar et al., 2015) to generate the configurations of our world and the agent interface for both training and testing phases. Additionally we also evaluate our trained agents on maps uniformly sampled from the $16 \times 16$ dataset originally used by Tamar et al. (2016), so as to get a direct comparison with the previous work, and to confirm the quality of our baseline. We tested all the models on the other available datasets ($8 \times 8$ and $28 \times 28$) too, without seeing significant changes in relative performance, so they are omitted from our evaluation.

Similar to Tamar et al. (2016), we employ a curriculum where the map size doesn't change at training time, but where the average length of the optimal path from the starting agent position is bounded by some value which gradually increases after a few training episodes. This allows the agent to more easily encounter the goal at initial stages of training, allowing for easier conditioning over the goal feature.

Across all our tests, both VProp and MVProp greatly outperformed our implementation of VIN (Table 1). Figure 2 shows rewards obtained during training, averaged across 5 training runs seeded

| maps | VIN | VProp | MVProp | VIN | VProp | MVProp |
|---|---|---|---|---|---|---|
| | | win rate | | | distance to optimal path | |
| $v$16x16 | $63.6\% \pm 13.2\%$ | $94.4\% \pm 5.6\%$ | 100% | $0.2 \pm 0.2$ | $0.2 \pm 0.2$ | $0.0 \pm .0$ |
| $32\times32$ | $15.6\% \pm 5.3\%$ | $68.8\% \pm 27.2\%$ | 100% | $0.8 \pm 0.3$ | $0.4 \pm 0.3$ | $0.0 \pm .0$ |
| $64\times64$ | $4.0\% \pm 4.1\%$ | $53.2\% \pm 31.8\%$ | 100% | $1.5 \pm 0.4$ | $0.5 \pm 0.4$ | $0.0 \pm .0$ |

Table 1: Average performance at the end of training of all tested models on the static grid-worlds with $90\%$ confidence value, across 5 different training runs (with random seeding). $v$16x16 correspond to the maps sampled from VIN's 16x16 grid test dataset, while the rest of the maps are sampled uniformly from our generator using the same parameters employed at training time. The distance to the optimal path is averaged only for successful episodes.

randomly. The original VIN architecture was mostly tested in a fully supervised setting (via imitation learning), where the best possible route was given to the network as target. In the appendix, however, Tamar et al. (2016) show that VIN can perform in a RL setting, obtaining an 82.5% success rate, versus the 99.3% success rate of the supervised setting on a map of $16 \times 16$. The authors do not show results for the larger $28 \times 28$ map dataset, nor do they provide learning curves and variance, however overall these results are consistent with the best performance we obtained from testing our implementation. That said, on average the model didn't perform as well as expected.

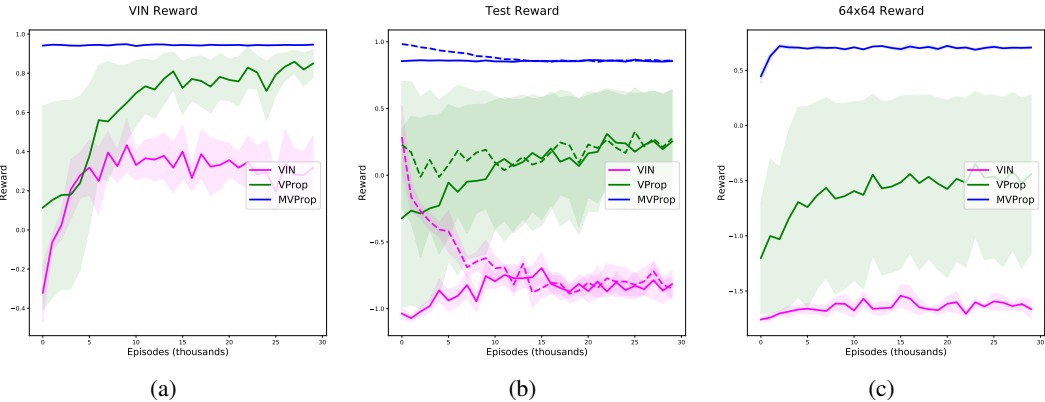

Figure 2: Average reward of all the models as they train on our curriculum. Note again that in both cases the map size is $32 \times 32$. a and c demonstrate performances respectively on the VIN dataset and our generated 64x64 maps. b shows performance on evaluation maps constrained by the curriculum settings (segmented line), and without (continuous line).

The final average performances of each model against the static-world experiments (Table 1) clearly demonstrate the strength of VProp and MVProp. In particular, MVProp very quickly learns the correctly approximate transition dynamics, resulting into strong generalization right from the first few hundreds episodes, hence obtaining near-optimal policies during the first thousand training episodes.

## 5.3 Tackling dynamic environments

Finally, we propose a set of experiments in which we allow our environment to spawn dynamic adversarial entities. Such entities at each step query a custom policy, which is executed in parallel to the agent's. Examples of these policies might include a $\epsilon$-*noop* strategy, which makes the entity move in random direction with probability $\epsilon$ or do nothing, or a $\epsilon$-direction policy, which makes the entity move to a specific direction with probability $\epsilon$ or do nothing. We use the first category of policies to augment our standard path-planning experiments, generating *enemies_only* environments where $20\%$ of the space is occupied by agents with $\epsilon = 0.5$, and *mixed* environments with the same amount of entities, half consisting of fixed walls, and the remaining of agents with $\epsilon = 0.2$ The latter type of policies is instead used to generate a deterministic but continuously changing environment which we call *avalanche*, in which the agent is tasked to reach the goal as quickly as possible while

avoiding "falling" entities (which also are also uniformly sampled up to $20\%$ of the walkable area). To deal with these new environments, the agent needs to re-plan at each step because the agent does

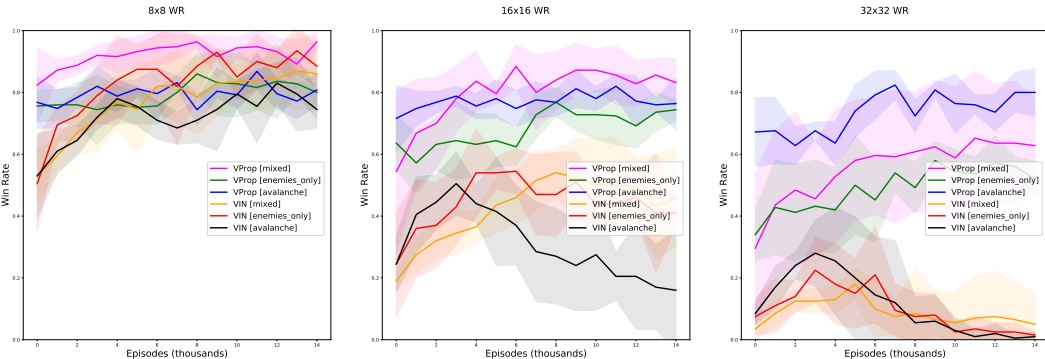

Figure 3: Average test win rate obtained on our stochastic experiments.

not have any prior knowledge about the entities policies, which forces us to train on 8x8 maps to reduce the time spent rolling-out the convolutional modules. This however allows us to train without curriculum, as the agent is more likely to successfully hit the goal in a smaller area. Figure 3 shows that VProp can successfully learn to handle the added stochasticity in the environment dynamics, and its generalization capabilities allow it to also successfully learn to tackle the *avalanche* maps on larger sizes (Figure 4). VIN agents however while managing to learn policies able to tackle small sizes, gradually lose performance on larger sizes, which become significantly harder to deal with unless the dynamics are modelled correctly.

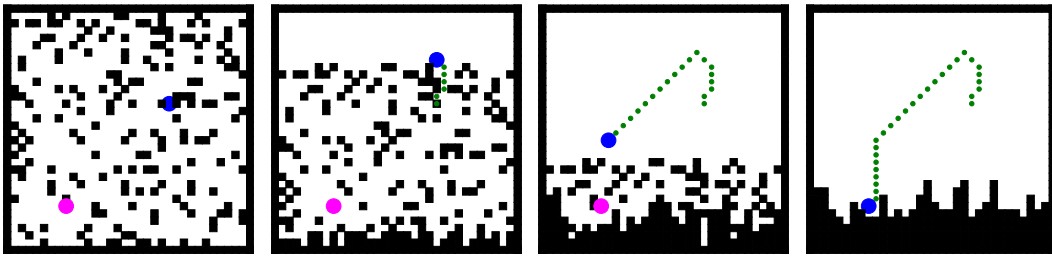

Figure 4: Example of policy obtained after training on a avalanche testing configuration. Agent and goal are shown as circles for better visualization, however they still occupy a single cell.

## 6 CONCLUSIONS

Architectures that try to solve the large but structured space of navigation tasks have much to benefit from employing planners that can be learnt from data, however these need to quickly adapt to local environment dynamics so that they can provide a flexible planning horizon without the need to collect new data and training again. Value Propagation modules' performances show that, if the problem is carefully formalized, such planners can be successfully learnt via Reinforcement Learning, and that great generalization capabilities can be expected when these models are built on convnets and are correctly applied to 2D path-planning tasks. Furthermore, we have demonstrated that our methods can even generalize when the environments are dynamics, enabling them to be employed in complex, interactive tasks. In future we expect to test our methods on a variety of tasks that can be embedded as graph-like structures (and for which we have the relevant convolutional operators). We also plan to evaluate the effects of plugging VProp into architectures that are employing VI modules (see Section 3), since most of these models could make use of the ability to propagate multiple channels to tackle more complex interactive environments. Finally, VProp architectures could be applied to algorithms used in mobile robotics and visual tracking (Lee et al., 2017; Bordallo et al., 2015), as they can learn to propagate arbitrary value functions and model a wide range of potential functions.

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
