# OpenReview forum: "Value Propagation Networks"
_ICLR.cc/2018/Conference — Invite to Workshop Track_

### Official Review · AnonReviewer2 · 2017-11-27
**An extension to value-iteration networks that improves performance on grid-worlds**

**Rating:** 5
**Confidence:** 4

**Review:**

The original value-iteration network paper assumed that it was trained on near-expert trajectories and used that information to learn a convolutional transition model that could be used to solve new problem instances effectively without further training.

This paper extends that work by
- training from reinforcement signals only, rather than near-expert trajectories
- making the transition model more state-depdendent
- scaling to larger problem domains by propagating reward values for navigational goals in a special way

The paper is fairly clear and these extensions are reasonable.  However, I just don't think the focus on 2D grid-based navigation has sufficient interest and impact.  It's true that the original VIN paper worked in a grid-navigation domain, but they also had a domain with a fairly different structure;  I believe they used the gridworld because it was a convenient initial test case, but not because of its inherent value.   So, making improvements to help solve grid-worlds better is not so motivating.  It may be possible to motivate and demonstrate the methods of this paper in other domains, however.  The work on dynamic environments was an interesting step:  it would have been interesting to see how the "models" learned for the dynamic environments differed from those for static environments.

---

> ### Author Response · Authors · 2018-01-03
> **Rebuttal for AnonReviewer2**
>
> Thank you for reviewing our work. We would like to address your comment about the relevancy of gridworlds as testbeds for our own work by providing three counter-arguments:
>
> - First and foremost, we have decided to focus on gridworlds because they are a largely used benchmark for work such as ours, and as such it allows to quickly compare methods. In particular, while Tamar et al. have indeed provided a variegated experimental section, their work has been almost entirely evaluated and re-used in experiments on gridworld or gridworld-like environments, which biased our experimental section towards making sure that such users would find it especially compelling. Sections 2 and 3 of our manuscript present some of such papers (e.g. [1], [2], [3]).
>
> - On all applications with a 2D structure of the original VIN paper, our method can be used as a drop-in replacement for the VI module. Whereas only experiments could confirm that our approach works on these domains as well, we believe our approach should indeed work on them (as the structure of the problem is always similar).
>
> - Finally, gridworld environments, while of simple construction and reasoning, can provide challenges that current algorithms are clearly unable to solve. We have for instance shown that when the environment becomes even slightly larger than sizes commonly used, state-of-the-art models struggle to learn and converge smoothly. We would like the community to take our work as inspiration and  try tackling gridworlds whose parameters (sizes, complexity of dynamics, sparsity of rewards, etc.) are pushed to areas that current algorithms cannot hope to solve. We for instance would like to reach a point where VProp we can tackle both _large_ and extremely _complex_ gridworlds, which would enable applied research to seriously consider it a planner that can be deployed in live systems.
>
>
> [1] https://arxiv.org/abs/1709.05273
> [2] https://arxiv.org/abs/1702.03920
> [3] https://arxiv.org/abs/1709.05706

---

### Official Review · AnonReviewer4 · 2017-11-27
**An extension of Value Iteration Network; the writing needs to be greatly improved**

**Rating:** 5
**Confidence:** 2

**Review:**

The paper introduces two alternatives to value iteration network (VIN) proposed by Tamar et al. VIN was proposed to tackle the task of learning to plan using as inputs a position and an image of the map of the environment. The authors propose two new updates value propagation (VProp) and max propagation (MVProp), which are roughly speaking additive and multiplicative versions of the update used in the Bellman-Ford algorithm for shortest path. The approaches are evaluated in grid worlds with and without other agents.

I had some difficulty to understand the paper because of its presentation and writing (see below).

In Tamar's work, a mapping from observation to reward is learned. It seems this is not the case for VProp and MVProp, given the gradient updates provided in p.5. As a consequence, those two methods need to take as input a new reward function for every new map. Is that correct?
I think this could explain the better experimental results

In the experimental part, the results for VIN are worse than those reported in Tamar et al.'s paper. Why did you use your own implementation of VIN and not Tamar et al.'s, which is publicly shared as far as I know?

I think the writing needs to be improved on the following points:
- The abstract doesn't fit well the content of the paper. For instance, "its variants" is confusing because there is only other variant to VProp. "Adversarial agents" is also misleading because those agents act like automata.

- The authors should recall more thoroughly and precisely the work of Tamar et al., on which their work is based to make the paper more self-contained, e.g., (1) is hardly understandable.

- The writing should be careful, e.g.,
value iteration is presented as a learning algorithm (which in my opinion is not)
\pi^* is defined as a distribution over state-action space and then \pi is defined as a function; ...

- The mathematical writing should be more rigorous, e.g.,
p.2:
T: s \to a \to s', \pi : s \to a
A denotes a set and its cardinal
In (1), shouldn't it be \Phi(o)? all the new terms should be explained
p. 3:
definition of T and R
shouldn't V_{ij}^k depend on Q_{aij}^k?
T_{::aij} should be defined
In the definition of h_{aij}, should \Phi and b be indexed by a?

- The typos and other issues should be fixed:
p. 3:
K iteration
with capable
p.4:
close 0
p.5:
our our
s^{t+1} should be defined like the other terms
"The state is represented by the coordinates of the agent and 2D environment observation" should appear much earlier in the paper.
"\pi_\theta described in the previous sections", notation \pi_\theta appears the first time here...
3x3 -> 3 \times 3
ofB
V_{\theta^t w^t}
p.6:
the the
Fig.2's caption:
What does "both cases" refer to? They are three models.
References:
et al.
YI WU

---

> ### Author Response · Authors · 2018-01-03
> **Rebuttal for AnonReviewer4**
>
> Thank you for reading our submission. Here's a response to the comments you made:
>
> - There is a single "reward map", as in Tamar et al. The reward used for the gradient update is that of the true task (e.g., -1 on hitting a wall, +1 on reaching the goal), not the reward map that is learnt.
>
> - As mentioned in Section 5.2, the best results we obtained from VIN _did_ match the numbers shown in the paper, however we saw a large variance in performance wrt random seeds when evaluated on many trials, even after some search on the RL hyperparameters.
> The original code release provided code for the supervised learning experiments, so it wasn’t applicable to our setup. In any case, we are confident our Pytorch implementation of the VIN model is essentially identical to the one in Theano provided by the authors, as it’s a relatively simple architecture and there are multiple similar implementations online.
>
> - Thank you for pointing out the typos in the abstract and the rest of the paper. These are going to be fixed in the version we will upload in a couple of days. We would like to point out that equation (1) has a typo and is hard to parse because of missing spaces (please, see the answer to reviewer 1). We appreciate the comment on clarity, and we will add a simpler explanation of VIN in the background section, which should help make the explanation of the baseline more readable.
>
> - Regarding your comments on the formalism regarding value iteration (and \pi), we will add a paragraph explaining that at training time we use stochastic policies, while testing with deterministic ones.
>
> - As far as we can see, the action set is usually denoted with the calligraphic letter, while the cardinal in standard uppercase when needed.
>
> - Further thanks for spotting the mistake in the definition of T (and R) on page 3. Please refer to our response to AnonReviewer1 (second point), we will correct the mistake.

---

### Official Review · AnonReviewer1 · 2017-11-27
**Useful extension to Value Iteration Networks to extend value of explicitly incorporate VI into problems with dynamic state**

**Rating:** 7
**Confidence:** 3

**Review:**

ORIGINALITY & SIGNIFICANCE

The authors build upon value iteration networks: the idea that the value function can be computed efficiently from rewards and transitions using a dedicated convolutional network. The authors point out that the original "value iteration network” (Tamar 2016) did not handle non-stationary dynamics models or variable size problems well and propose a new formulation to extend the model to this case which they call a value propagation network.  It seems useful and practical to compute value iteration explicitly as this will propagate values for us without having to learn the propagated form through extensive gradient update steps. Extending to the scenario of non-stationary dynamics is important to make the idea applicable to common problems. The work is therefore original and significant.

The algorithm is evaluated on the original obstacle grids from Tamar 2016 and larger grids generated to test scalability. The authors Prop and MVProp are able to solve the grids with much higher reliability at the end of training and converge much faster.  The M in MVProp in particular seems to be very useful in scaling up to the large grids. The authors also show that the algorithm handles non-stationary dynamics in an avalanche task where obstacles can fall over time.


QUALITY

The symbol d_{rew} is never defined — what does “new” stand for? It appears to be the number of latent convolutional filters or channels generated by the state embedding network.

Section 2.2 Sentence 2: The final layer representing the encoding is given as ( R^{d_rew  x d_x x d_y }.
Based on the description  in the first paragraph of section 2, it sounds like d_rew might be the number of channels or filters in the last convolutional layer.

In equation 1, it wasn’t obvious to me that the expression max_a q_{ij}^{k-1} q^{k} corresponds to an actual operation?
The h( \Phi( x ), v^{k-1} ) sort of makes sense …  value is only calculated with respect to only the observation of the maze obstacles but the policy \pi is calculated with respect to the joint  observation and agent state.

The expression

   h_{aid} ( \phi(0), v )   =   <  Wa,   [ \phi(o) ; v ]   >   +   b

makes sense and reminds me of the Value Iteration network work where we take the previous value function, combine it with the reward function and use convolution to compute the expectation (the weights Wa encode the effect of transitions). I gather the tensor Wa = R^{|A| x (d_{rew} x d_x x d_y } both converts the feature embedding \phi{o} to rewards and represents the transition / propagation of reward across states due to transitions and discounts at the same time?

I didn’t understand the r^in, r&out representation in section 4.1. These are given by the domain?

I did get the overall idea of efficiently creating a local value function in the neighborhood of the current state and passing this to the policy so that it can make a local decision.

A bit more detail defining terms, explaining their intuitive role and how the output of one module feeds into the next would be helpful.


POST REVISION COMMENTS:

- I didn't reread the whole thing -  just used the diff tool.
- It looks like the typos in the equations got fixed
- The new phrase "enables to learn to plan" seems pretty awkward

---

> ### Author Response · Authors · 2018-01-03
> **Rebuttal for AnonReviewer1**
>
> Thank you for reading and reviewing our work. We really appreciate the comments on novelty and significance.
>
> - As you indeed spotted, d_rew definition is implied in Section 2, where it indicates the number of feature channels extracted by a embedding function in the input. We used it to refer to Tamar et al. ‘16, but we agree that it’s a bit confusing if you don’t carefully read the section. We’ll rename it to d_feat.
>
> - Equation (1) is a literal translation of the paragraph above it (even though there is a typo). It is hard to parse because there are missing spaces between q^{k-1}_{aij} and q^k, so the reader can’t see there are two equalities; also we will make it clear that v^k depends on q^k and not q^{k-1}. That max operation in equation (1) is formalization of the max-pooling operation performed by the convnet at each iteration of k. In the case of VI it so happens that the operation is performed over the set of actions A, and it’s useful to point it out to the reader to provide a summary of the value iteration -> VI module mapping.
>
> - W_a indeed computes the transition map when d_rew := A and \phi(o) := R. This particular formulation is useful when implementing a VI module, as it provides the dimensions for the module when using a single fully-connected linear layer to represent the transform.
>
> - Thank you for spotting the missing definition. r_in and r_out are the reward propagation maps that can be generated by reparametrizing the single VI reward map. They are properly defined and used in the following paragraph to define VProp’s value recurrence, but we’ll add a quick explanation where they are first mentioned.

---

### Decision · Program_Chairs · 2018-01-29
**ICLR 2018 Conference Acceptance Decision**

**Decision:**

Invite to Workshop Track

**Comment:**

This paper and reviews makes for a difficult call.  The reviewers appear to be in agreement that Value Propagation provides an interesting algorithmic advance over earlier work on Value Iteration networks.  AnonReviewer1 gives a strong rationale why the advance is both original and significant.  Their experiments also show very nice results with VProp and MVProp in 2-D grid-worlds.

However, I also fully agree with AnonReviewer2 that testing in other domains beyond 2-D grid-world is necessary.  Earlier work on VIN was also tested on a Mars Rover / continuous control domain, as well as graph-based web navigation task.  The authors' rebuttal on this point comes across as weak.  In their view, they can't tackle real-world domains until VProp has been proven effective in large, complex grid-worlds.  I don't buy this at all -- they could start initial experiments right away, which would perhaps yield some surprising results. Given this analysis, the committee recomments this paper for workshop.

Pros: significant algorithmic advance, good technical quality and writeup, nice results in 2-D grid world.

Con: Validation is only in 2-D grid-world domains.